# Ca$_2$Fe$_2$O$_5$-Based WGS Catalysts to Enhance the H$_2$ Yield of Producer Gases

**Isabel Antunes** [1,*] [ID]**, Luís C. M. Ruivo** [1,2] [ID]**, Luís A. C. Tarelho** [2] [ID] **and Jorge R. Frade** [1,*]

[1]   Department of Materials and Ceramic Engineering, CICECO—Aveiro Institute of Materials,
     University of Aveiro, 3810-193 Aveiro, Portugal; luis.ruivo@ua.pt
[2]   Department of Environment and Planning, CESAM—Centre for Environmental and Marine Studies,
     University of Aveiro, 3810-193 Aveiro, Portugal; ltarelho@ua.pt
*   Correspondence: isabel.antunes@ua.pt (I.A.); jfrade@ua.pt (J.R.F.)

**Abstract:** Ca$_2$Fe$_2$O$_5$-based catalysts were synthesized from siderite and calcite precursors, which were processed in the form of pelletized samples and tested as water gas shift catalysts. Catalytic tests were performed in a tubular reactor, at temperatures in the range 400–500 °C and with different H$_2$O:CO ratios, diluted with N$_2$; this demonstrates the positive impact of Ca$_2$Fe$_2$O$_5$ on conversion of CO and H$_2$ yield, relative to corresponding tests without catalyst. The catalytic performance was also remarkably boosted in a microwave-heated reactor, relative to conventional electric heating. Post-mortem analysis of spent catalysts showed significant XRD reflections of spinel phases (Fe$_3$O$_4$ and CaFe$_2$O$_4$), and SiO$_2$ from the siderite precursor. Traces of calcium carbonate were also identified, and FTIR analysis revealed relevant bands ascribed to calcium carbonate and adsorbed CO$_2$. Thermodynamic modelling was performed to assess the redox tolerance of Ca$_2$Fe$_2$O$_5$-based catalysts in conditions expected for gasification of biomass and thermochemical conditions at somewhat lower temperatures (≤500 °C), as a guideline for suitable conditions for water gas shift. This modelling, combined with the results of catalytic tests and post-mortem analysis of spent catalysts, indicated that the O$_2$ and CO$_2$ storage ability of Ca$_2$Fe$_2$O$_5$ contributes to its catalytic activity, suggesting prospects to enhance the H$_2$ content of producer gases by water gas shift.

**Keywords:** water gas shift; brownmillerite; H$_2$ production; microwave heating; thermodynamic modelling

## 1. Introduction

Thermal conversion of lignocellulosic biomass by gasification can be an important contribution to the carbon-neutral economy since producer gas mixtures may be used as commodities for other industrial applications. Though the biomass-derived gas is greatly influenced by process conditions, the typical H$_2$:CO molar ratio is often lower than 1:1 [1], even after biomass steam gasification [2]. Consequently, depending on the end-use application (i.e., methanation), the contents of H$_2$ in the producer gas must be upgraded by water gas shift (WGS) reaction. The WGS reaction may be performed at relatively low (LTS, 150–300 °C) or intermediate (HTS, 350–500 °C) temperatures, relying on a wide diversity of catalysts or non-catalytic processes in less common environments, such as supercritical water or plasma [3], or unusual conditions, such as microwave irradiation [4]. Classical WGS catalysts for LTS and HTS processes are based on Fe/Cr and Cu/Zn mixed oxides [5,6]. Still, the environmental impact of Cr content raised concerns; this stimulated research on Cr-free catalysts with incorporation of other transition metal oxides and lanthanides [7], or even noble metals. Ni-based catalysts are also affected by limitations such as promotion of methanation [8]. Moreover, Cu/Zn based-catalysts are readily deactivated at low sulphur content (<0.5 ppm), which requires previous cleaning of the biomass-derived gas [9]. Thus, alternative low-cost catalysts are still needed for specific applications such as H$_2$-enriched producer gas by a combination of biomass gasification and WGS [10], or supercritical water

gasification [11], possibly operating also in less common conditions. An interesting option is based on the application of Fe/Ca-based materials, due to their thermal stability, low cost and effective activity towards WGS and reforming reactions.

$Ca_2Fe_2O_5$-based catalysts are attracting attention for their activity in biomass gasification [12], including chemical looping gasification [13], and relevant mechanisms which enhance the yield of $H_2$ in the producer gas and promote tar conversion by steam reforming [14]. The catalytic activity has also been demonstrated for a variety of other processes, including catalysts for steam reforming of methane [15], chemical looping gasification of coal [16], or chemical looping combustion [17], decomposition of $NO_x$ in exhaust gases [18], catalyst supports for oxidation of CO [19], etc. These catalysts are based on low-cost calcium and iron oxides ($Fe_xO_y$), and their synthesis can be achieved from low-cost precursors at relatively low temperatures [20].

Gasification of biomass assisted with co-additions of iron and calcium oxides yielded a slight increase in gasification efficiency and enhanced the yield of $H_2$ [21]; this can be related to redox cycles in the presence of $CO/CO_2$ and $H_2O/H_2$ pairs [22] and was ascribed to a combination of chemical looping provided by $Fe_xO_y$ and adsorption of $CO_2$ by CaO. Similarly, enhanced gas yield was reported for steam gasification of coal catalyzed by calcium ferrites with different Ca:Fe ratios [16]. The yield of $H_2$ reached a maximum at the intermediate Ca:Fe ratio, which was ascribed to promotion of the water gas shift (WGS) reaction by $Ca_2Fe_2O_5$.

One also expects good sulphur tolerance of calcium ferrites in contact with producer gases from biomass or other low-grade energy sources, based on the demonstrated ability to capture $H_2S$ and other contaminants [23]. The oxygen looping can be related to variable oxygen sub-stoichiometry and the rich structural diversity of iron species with different oxidation states and co-existing octahedral and tetrahedral coordination [24]. The brownmillerite structure of $Ca_2Fe_2O_5$ is also highly stable in wide ranges of redox conditions, except for the onset of traces of CaO [25], exceeding the stability of $Fe_xO_y$ and also $CaFe_2O_4$ [24].

The promotion of $H_2$ yield in producer gas obtained by biomass gasification [26] has been interpreted as a promotion of the WGS reaction (Equation (1)) in gas mixtures containing high contents of CO and steam:

$$CO + H_2O \rightarrow CO_2 + H_2; \tag{1}$$

Still, a different work suggests that $Ca_2Fe_2O_5$ may also promote the reverse WGS reaction [27] in $CO_2$- and $H_2$-rich gas mixtures; this indicates that the catalyst promotes convergence to equilibrium (Equation (2)) from both sides.

$$k_{eq} = \frac{pCO_2 pH_2}{pCO\ pH_2O}. \tag{2}$$

Note that the equilibrium constant of WGS corresponds to a difference between the redox conditions of the $H_2/H_2O$ and $CO/CO_2$ redox pairs; this depends on absolute temperature (Equation (3)) [28], and is reverted at about 800 °C.

$$ln\left(\frac{pH_2}{pH_2O}\right) - ln\left(\frac{pCO}{pCO_2}\right) = ln(k_{eq}) \tag{3}$$

$$\approx -13.15 + 5.44 \times 10^{-4}T - 1.125 \times 10^{-7}T^2 + 1.077\ln(T) + \frac{5.694 \times 10^3}{T} - \frac{4.917 \times 10^4}{T^2}.$$

On the other hand, the operation of conventional WGS reactors at high pressures also notably penalizes the economic feasibility of the gasification plant [29]. Note that WGS processes are usually carried out in fixed-bed reactors, and truly isothermal conditions are seldom reached in the packed bed. As a result, high energy inputs are involved to balance heat transfer limitations, which points out the necessity to explore alternative energy-

efficient methods. In this perspective, microwave-assisted operation can significantly improve the performance of WGS catalysts. Compared with conventional heating, where heat is shifted from the surface to the core of the material through conduction driven by temperature gradients, microwaves induce local heating by direct conversion of the electromagnetic field into heat; this promotes direct heating of catalyst particles.

Therefore, the present work was intended to confirm the catalytic activity of $Ca_2Fe_2O_5$ to promote the WGS reaction, at atmospheric pressure while suppressing methanation, and to assess the impact of microwave irradiation on catalytic performance. The $Ca_2Fe_2O_5$-based catalyst was selected for its Fe- and Ca-based composition, structural stability in wide redox ranges, prospective economic feasibility based on abundant elements (Ca and Fe), and ability to be processed from low-cost precursors without previous separation of gang components, by a facile method [20].

One seeks enhancement of the $H_2$ yield of producer gas by secondary treatment based on WGS, after a primary step of biomass steam gasification. Thus, working conditions for WGS treatment were focused on a relatively high temperature range (400–500 °C) to prevent condensation of tars. The range of the steam to carbon monoxide ratio $H_2O:CO$ cannot be directly taken from the reported composition of producer gas, which is measured after condensation of steam and, thus, reported on a dry basis [1]. Nevertheless, one predicts approximate values for the concentration of steam and $H_2O:CO$ in the producer gas before condensation, by combining typical values of the H:C elemental ratio in the biomass feedstock $(H:C)_{biom} \approx 1.5$ [1] with the additional contribution of the steam:carbon ratio added to assist gasification $(S:C)_{gas} \geq 0.5$ [2]; this yields the effective ratio $(H:C)_{eff} \approx (H:C)_{biom}\left\{1 + 2(S:C)_{gas}\right\} \geq 2.5$, and on combining with elemental balances of H and C in reported producer gas compositions (e.g., ref. [2]), one expects $H_2O:CO$ in the range 1–2 depending on temperature and other operating conditions.

## 2. Results and Discussion

### 2.1. Catalytic Testing

WGS catalytic testing was performed in a microwave reactor or in a conventional electric furnace, and blank tests were also performed without a catalytic bed. Other operating conditions were identical for every case, namely temperature (500 °C), $H_2O:CO$ feed ratio ($H_2O:CO = 3$) and gas hourly velocity (GHSV = 5900–6750 $h^{-1}$). Gas analysis only detected $H_2$, CO and $CO_2$, without any traces of light hydrocarbons ($CH_4$, $C_2H_6$, etc.) or other volatile compounds, and Table 1 summarizes the relevant results in terms of conversion of CO and $H_2$ yield, relative to total contents of C-containing gases. Blank results without a catalyst did not show significant differences with and without microwaves, whereas the combination of catalyst with microwave heating yielded higher conversion of CO to $CO_2$ and also enhanced $H_2$ yield.

**Table 1.** CO conversion to $CO_2$ and corresponding yield of $H_2$ obtained by water gas shift reaction at 500 °C, under feed ratio $H_2O:CO = 3$, without catalysts (blank) and with the $Ca_2Fe_2O_5$-based catalyst, performed with conventional electric heating or microwave heating.

| Heating Type | Yield (%) | Blank | Catalyst |
|---|---|---|---|
| Conventional | $CO_2$ | 0.2 | 17 |
| Microwave | | 0.4 | 61 |
| Conventional | $H_2$ | 0.1 | 16 |
| Microwave | | 0.2 | 58 |

Thus, we found clear evidence of catalytic activity of $Ca_2Fe_2O_5$ and remarkable microwave boosting, possibly related to the microwave absorption properties of ferrite-based catalysts, and prospects for direct self-heating under microwave irradiation. Note that although the WGS reaction is moderately exothermic, this may be insufficient to account for the sensible heat of reactants from room temperature up to the reaction temperature,

which increases with the reaction temperature and with the $H_2O:CO$ feed ratio [3]. In fact, microwave-assisted WGS was previously proposed with a commercial Fe–Cr-based catalyst at intermediate temperatures [4] and with Cu-Zn catalyst at lower temperatures [30]; this work was also supported by modelling, which revealed significant temperature differences, with overheating in dense parts of the reactor. Other references also analyzed the applicability and limitations of microwave reactors for other processes intended for catalytic valorization of biomass-derived products [31].

The yields of $H_2$ and $CO_2$ in Table 1 are similar, within the range of expected experimental errors, and indicate that the WGS reaction (Equation (1)) prevails. Slight differences between the yields of $H_2$ and $CO_2$ might still be ascribed to direct oxidation ($CO + 0.5 \cdot O_2 \rightarrow CO_2$) [22], relying on the oxygen storage ability of the catalyst by phase transformations or by variable oxygen stoichiometry of calcium ferrites [25]. In addition, the $CO/CO_2$ balance may be slightly changed by the onset of calcium carbonate ($CaCO_3$), as revealed by post-mortem analysis.

In fact, X-ray diffraction showed significant differences between the as-prepared catalyst and post-mortem analysis of spent catalysts (Figure 1). The as-prepared catalyst contained mainly the $Ca_2Fe_2O_5$-based brownmillerite phase (JCPDS 00-047-1744), with $CaFe_2O_4$ (JCPDS 01-072-1199) as secondary phase, unreacted $SiO_2$ (JCPDS 00-005-0490) from the natural siderite precursor and traces of hematite $Fe_2O_3$ (JCPDS 00-033-0664). The most relevant changes in spent catalyst samples refer to extinction of the $Fe_2O_3$ phase, onset of magnetite $Fe_3O_4$ (JCPDS 01-088-0866) and also $CaCO_3$ (JCPDS 00-005-0586). Note that its main reflection (104) at $\approx 29.41°$ (Figure 1) is significantly shifted from the (131) reflection of $Ca_2Fe_2O_5$ ($\approx 29.22°$).

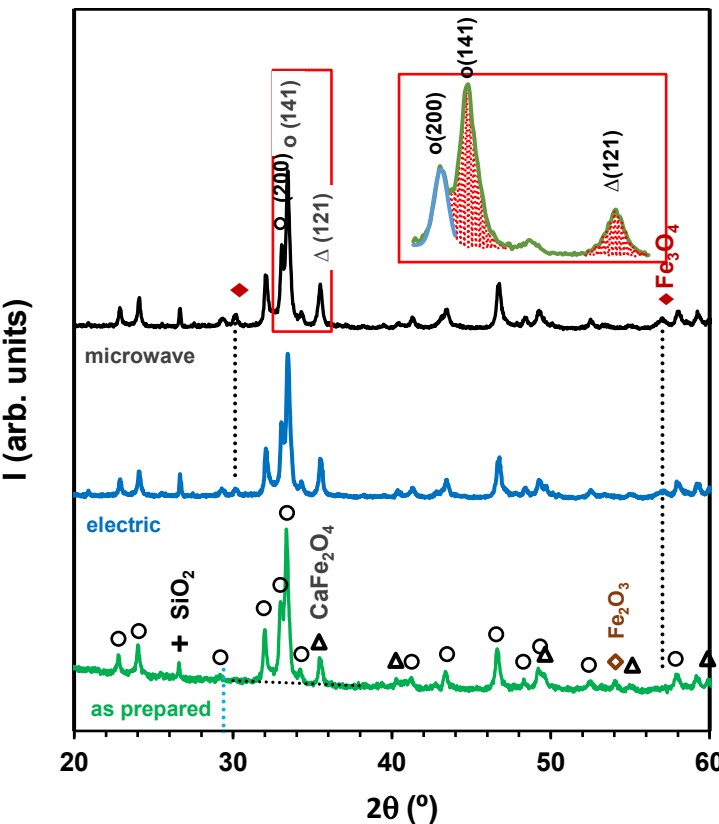

**Figure 1.** X-ray diffractograms of as-prepared catalyst samples and spent catalysts after water gas shift testing at 500 °C, with $H_2O:CO$ ratio = 3:1 and GHSV = 5900–6750 $h^{-1}$, in conventional electric furnace or microwave heating. The markers identify reflections ascribed to $Ca_2Fe_2O_5$ (○), (△) $CaFe_2O_4$, (+) $SiO_2$, (◇) $Fe_2O_3$ and (◆) $Fe_3O_4$.

The relative integrated intensity of the main reflection of the secondary phase $I_{\Delta(121)}$ : $I_{\bigcirc(141)}$ is also somewhat higher in spent catalyst samples, where $I_{\bigcirc(141)}$ and $I_{\Delta(121)}$ denote the integrated intensities of the main reflections of $Ca_2Fe_2O_5$ and $CaFe_2O_4$. The integrated intensity of the (141) reflection of $Ca_2Fe_2O_5$ was calculated after de-convolution of the partially superimposed (200) and (141) reflections, as shown by the insert in Figure 1. The integrated intensity ratio $I_{\Delta(121)}$ : $I_{\bigcirc(141)}$ increased from 0.19 in the as-prepared catalyst to 0.30 after catalytic testing with microwave heating. The integrated intensity ratio was also slightly higher after catalytic testing with microwave heating (0.30) than after testing in the electric furnace (0.25).

A more-detailed study of the catalytic activity of the $Ca_2Fe_2O_5$-based catalyst was performed to assess the dependence of conversion of CO and yield of $H_2$ on temperature and $H_2O$:CO feed ratio. Temperature plays a key role in the conversion of CO, which can be ascribed mainly to kinetic limitations, since thermodynamic equilibrium should reach about 93% at 500 °C or 99% at 400 °C. we also assessed a representative rate constant (k) based on deviation from thermodynamic equilibrium, combined with the average value of GHSV $\approx 6325$ h$^{-1}$, i.e.,

$$k = GHSV\{1 - x_{CO}/x_{CO,eq}\} \approx 15.7 \exp(-3919/T). \tag{4}$$

The corresponding activation energy ($E_a \approx 33$ kJ/mol) was close to the lowest values reported for WGS with Pt/FeO$_x$ catalysts [32], and also for WGS reaction with Cu/Zn-based catalysts under microwave heating [30]. However, this similarity was inconclusive since the literature data are highly scattered and may depend on the temperature range [3].

The results of thermodynamic equilibrium can be ascribed to a combination of WGS and methanation (Equation (5)), with a corresponding decrease in yield of $H_2$ and changes in the stoichiometric $H_2$:$CO_2$ ratio, mainly at lower temperatures:

$$CO + (1 - 2x)H_2O \rightarrow (1 - x)CO_2 + xCH_4 + (1 - 4x)H_2 \tag{5}$$

However, $CH_4$ was not detected in the actual experimental results (Figure 2), indicating that the catalysts hinders the reaction of methanation while promoting WGS. Thus, the effective yield of $H_2$ is still close to equilibrium and in the same range as the yield of $CO_2$, as expected for the prevailing WGS (Equation (1)). Note that the side reaction of methanation is considered a drawback of some WGS catalysts, such as Ni-based catalyst, and suitable changes in catalyst composition are needed to minimize the negative impact on $H_2$ yield [33].

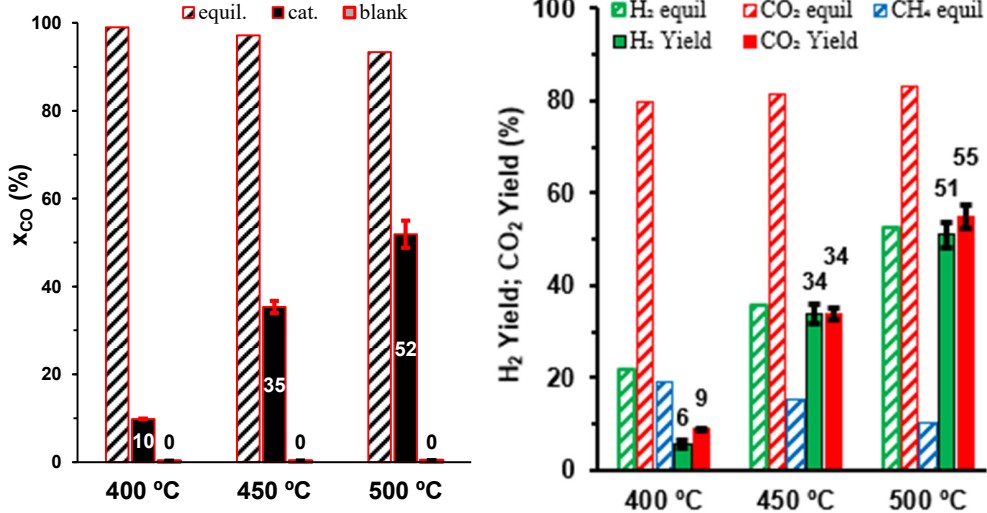

**Figure 2.** Temperature dependence of CO conversion and yields of $H_2$, $CO_2$ and $CH_4$ in equilibrium and corresponding experimental results for feed ratio $H_2O$:CO = 2 and GHSV = 5900–6750 h$^{-1}$.

Thermodynamic predictions also indicated that the impact of the side reaction of methanation should decrease with increasing $H_2O$:CO ratio (Figure 3). However, the corresponding experimental results only showed a slight gain in the yield of $H_2$ for an intermediate steam:CO ratio, and this trend was reverted for the highest steam:CO ratio. In addition, the yields of $H_2$ and $CO_2$ remained similar, indicating that methanation remains negligible independently of the steam:CO ratio. Thus, an excessive $H_2O$:CO ratio affects the WGS reaction (Equation (1)), as revealed by increasing differences between the experimental yield of $H_2$ and the corresponding equilibrium values (Figure 3).

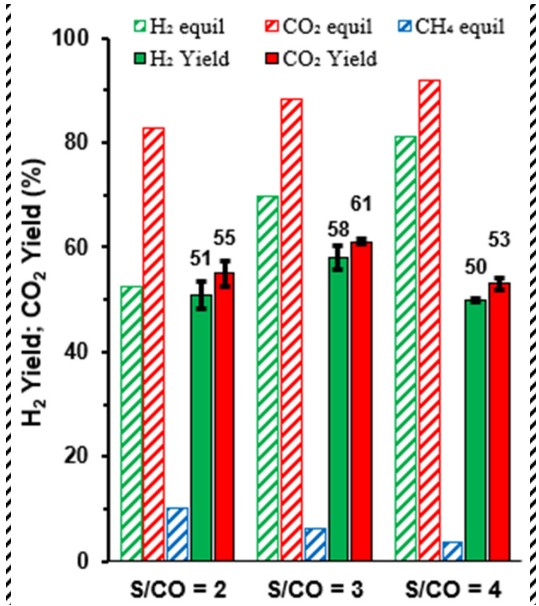

**Figure 3.** Temperature dependence of yields of $H_2$, $CO_2$ and $CH_4$ in equilibrium and corresponding experimental results vs. the feed ratio $H_2O$:CO (S/CO), at 500 °C and for GHSV = 5900–6750 $h^{-1}$.

### 2.2. Post-Mortem Analysis

Post-mortem analysis of spent samples (Figure 4) showed that the reacting gas mixture exerted significant effects on the catalyst, with emphasis on onset of $CaCO_3$ and $Fe_3O_4$. The main reflection of $CaCO_3$ (104) increased with the temperature of the catalytic tests, in close relation with increasing catalytic activity (Figure 2). In addition, the highest $H_2O$:CO ratio gave rise to aragonite, indicating that excessive humidity promotes ready $CO_2$ adsorption and/or carbonation, with a negative impact on catalytic performance (Figure 3), possibly by blocking active sites. In fact, humidity often assists adsorption of $CO_2$ by capture materials [34] and promotes readier carbonation of CaO-rich materials such as cements. The intensities of $Fe_3O_4$ reflections also increased with the temperature of catalytic testing, by gradual reduction of traces of $Fe_2O_3$, possibly combined with additional segregation of $Fe_3O_4$ at onset of $CaCO_3$.

Post-mortem FTIR analyses (Figure 5) confirmed the onset of calcium carbonate, revealed by asymmetric stretching of carbonate groups at 1490–1420 $cm^{-1}$ [35], combined with adsorbed $CO_2$, revealed by the band at 2360–2330 $cm^{-1}$, by analogy with catalysts impregnated with alkaline earth oxides [36]. Note that the relative amplitudes of corresponding bands have been adjusted, taking the Fe-O stretching band ($\approx$570 $cm^{-1}$) [37] as reference. Thus, both processes increased mainly with the temperature of the catalytic tests. The $H_2O$:CO ratio also determined mainly the adsorption band, suggesting combined effects of humidity and $CO_2$, as reported for carbon capture materials [34,38]. On the contrary, the impact of steam:CO on the carbonate band was far from clear, since it decayed at intermediate $H_2O$:CO and reverted for the highest steam:CO value. Note also that $CO_2$ adsorption may also have occurred preferentially on cooling to room temperature, after the catalytic tests. In fact, the as-prepared catalyst showed a strong band ascribed to adsorbed

CO$_2$, possibly because the initial firing temperature (800 °C) may have activated basic sites for subsequent adsorption of CO$_2$ at room temperature.

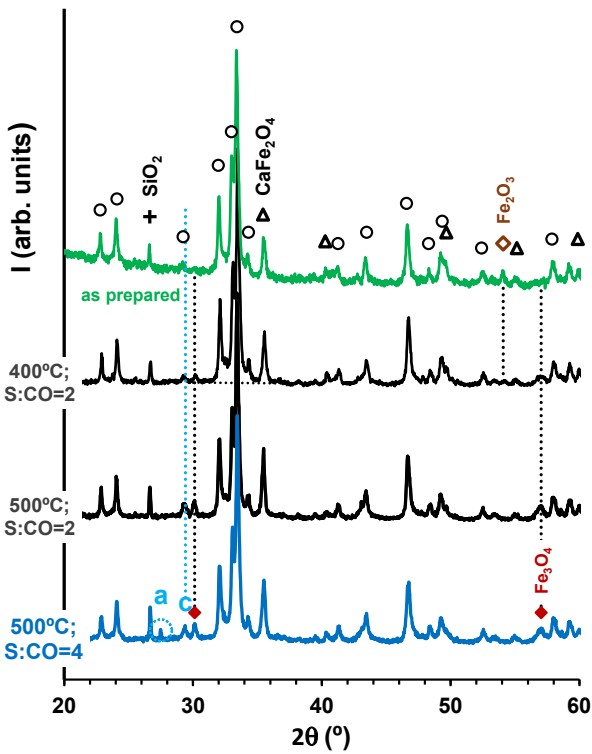

**Figure 4.** X-ray diffractograms of as-prepared catalyst samples and spent catalysts after water gas shift testing under different combinations of temperature and H$_2$O:CO ratio (S/CO), with microwave irradiation. The markers identify reflections ascribed to Ca$_2$Fe$_2$O$_5$ (○), (△) CaFe$_2$O$_4$, (+) SiO$_2$, (◇) Fe$_2$O$_3$ and (◆) Fe$_3$O$_4$. The calcite and aragonite, polymorphs of CaCO$_3$ are identified by (c) and (a).

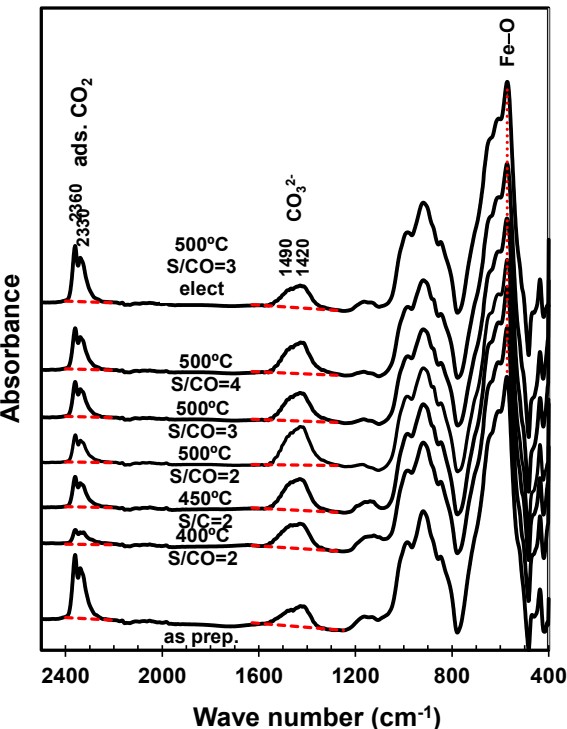

**Figure 5.** FTIR spectra of the as-prepared catalysts and spent catalyst samples tested at different combinations of temperature and H$_2$O:CO ratio (S/CO).

Scanning electron microstructures of spent catalysts (Figure 6) confirmed the heterogeneous features of the catalyst samples, with relatively coarse crystals of gang components ($SiO_2$) from the siderite precursor. Low magnification microstructures of spent catalysts did not show any evidence of onset of fractures, mechanical failure or significant erosion. On the other hand, higher magnification was still ill-suited to obtain precise assessment of grain sizes for the main phase ($Ca_2Fe_2O_5$), except possibly for a crude range $\leq 1$ µm.

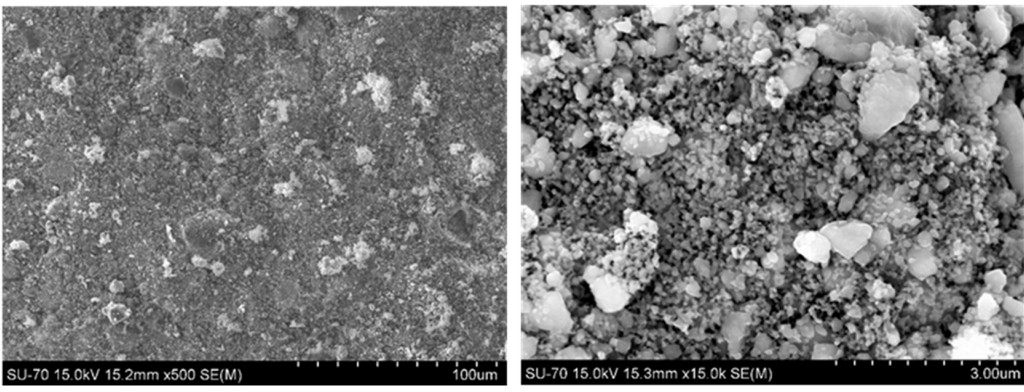

**Figure 6.** Scanning electron microstructures of a spent catalyst after testing at 500 °C, with feed ratio $H_2O:CO = 2$, under microwave irradiation.

X-ray diffractograms confirmed nanostructuring of the main brownmillerite phase $Ca_2Fe_2O_5$ by estimating an approximate range for the crystallite size $D$ from full width half maximum $\beta$ and $\theta$ angles of the main X-ray reflections (Figures 1 and 4), as described by the Debye–Scherrer equation [39]:

$$D \approx \lambda / \{\beta cos(\theta)\} \tag{6}$$

this yielded typical sizes in the order of $36 \pm 5$ nm for as-fired samples, and similar results for spent samples, after catalytic testing at temperatures, namely $39 \pm 5$ nm at 400 °C and $37 \pm 4$ nm at 500 °C. On combining this crystallite size range with the density of the main brownmillerite phase ($\rho \approx 3.70$ g/cm$^3$), we also estimated typical values of specific surface area, in the order of $S \approx 3/(D\rho) \approx 20$ m$^2$/g.

### 2.3. Thermodynamic Guidelines

Figure 7 shows thermodynamic predictions for the Ca−Fe−O−C system at the firing temperature of the catalyst (800 °C), as a guideline for synthesis from the mixture of carbonates. On assuming complete reactivity of a stoichiometric mixture of carbonate precursors, while neglecting impurities in the low-grade siderite precursors and assuming also a self-controlled redox condition by evolving $CO/CO_2$ gas mixtures:

$$2CaCO_3 + 2FeCO_3 \rightarrow Ca_2Fe_2O_5 + CO + 3CO_2 \tag{7}$$

Thus, the redox conditions in a closed atmosphere should be along the **a**–**b** line in Figure 7. However, the X-ray diffractogram of the as-fired catalyst sample (Figure 1) showed co-existence of the $Ca_2Fe_2O_5$- and $CaFe_2O_4$-based phases, combined with segregation of $Fe_2O_3$ and $SiO_2$ from the low-grade siderite precursor; this indicated decomposition of the brownmillerite phase, probably induced by gradual evolution towards oxidizing conditions, through the intermediate ternary point $Ca_2Fe_2O_5/CaFe_2O_4/Fe_3O_4$ (**c** in Figure 7), and then oxidation of the $Fe_3O_4$ fraction to $Fe_2O_3$ (**d** in Figure 7). Note that the apparent deviation from the stoichiometric Ca:Fe = 2 ratio may be explained by incorporation of light elements (Mg and Al from the siderite precursor) in the brownmillerite phase, i.e., $Ca_2(Fe,Al,Mg)_2O_5$ [40], with impact on relevant properties such as magnetization and coercivity [41].

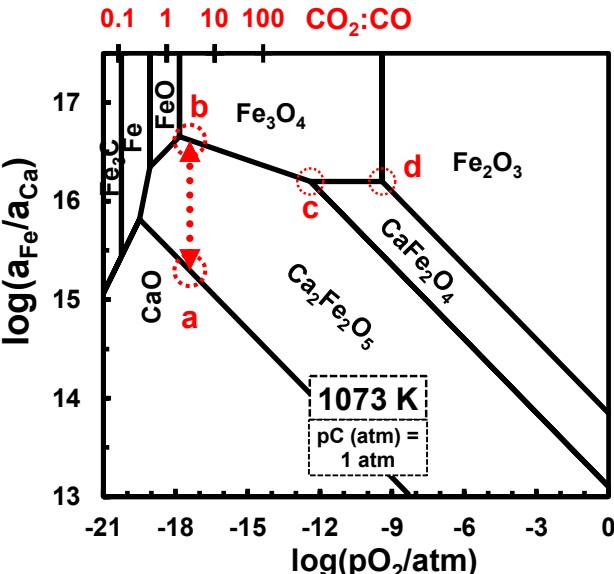

**Figure 7.** Phase stability for the Ca−Fe−O−C diagram vs. oxygen partial pressure, at 800 °C and in equilibrium with CO/CO$_2$ atmospheres, under $p_C = pCO + pCO_2 = 1$ *atm*. The secondary horizontal axis shows the corresponding scale for the CO$_2$:CO ratio.

Thermodynamic predictions for the Ca−Fe−O−C system and interactions between catalyst and reacting gas mixture at 400 °C are shown in Figure 8, for gas feed diluted in N$_2$, with $p_C = pCO + pCO_2 \approx 0.03$ atm and $\approx 0.007$ atm. In the case of $p_C = pCO + pCO_2 \approx 0.03$ atm (Figure 8/top), the equilibrium redox condition for the feed ratio H$_2$O:CO = 2 (vertical dotted line) was outside the upper limit of the redox stability range of the Ca$_2$Fe$_2$O$_5$-based brownmillerite phase. On the contrary, the experimental results showed low conversion of CO, retaining a very low CO$_2$:CO ratio, below the low limits of the stability range of Ca$_2$Fe$_2$O$_5$. Though this suggests risks of carbon deposition, we did not find any traces of carbon deposits, probably because the effective phase composition of the catalyst (Figure 4) did not include the most active metallic phase. In addition, microwave heating combined with CaO-rich catalysts may promote the highly endothermic Boudouard reaction ($C + CO_2 \rightarrow 2CO$) [42].

On combining experimental evidence of catalyst phase composition (Figure 4) with thermodynamic predictions (Figure 8), we may also assume that onset of the calcium carbonate and magnetite was consistent, with reactivity with mixtures of CO and CO$_2$:

$$Ca_2Fe_2O_5 + 1/3CO + 5/3CO_2 \rightarrow 2/3Fe_3O_4 + 2CaCO_3; \tag{8}$$

In this case, carbonation may have contributed to maintaining a low CO$_2$:CO ratio after catalytic testing at 400 °C, by preferential consumption of CO$_2$.

The effective phase composition of the spent catalyst after testing at 400 °C (Figure 4) also indicated co-existence of the Ca$_2$Fe$_2$O$_5$-based brownmillerite phase with the main CaFe$_2$O$_4$-based secondary phase, and also weaker reflections of Fe$_3$O$_4$ and CaCO$_3$; this differed from the predicted equilibrium conditions, which suggested extensive carbonation and the absence of CaFe$_2$O$_4$ for the actual experimental gas feed with $p_C = pCO + pCO_2 \approx 0.03$ atm (Figure 8/top). Actually, the effective phase composition of the spent catalyst was more consistent with non-equilibrium conditions, evolving by non-uniform redox conditions, as revealed by residual traces of Fe$_2$O$_3$, and only incipient carbonation; this is consistent with the equilibrium phase diagram for lower $pCO + pCO_2$ contents, as predicted for $pCO + pCO_2 \approx 0.007$ atm (Figure 8/bottom). In this case, the phase stability diagram predicted separate 3-phase contacts for Ca$_2$Fe$_2$O$_5$/CaCO$_3$/Fe$_3$O$_4$

(**a** in Figure 8/bottom) and for $CaCO_3/CaFe_2O_4/Fe_3O_4$ (**b** in Figure 8/bottom). In the second case, carbonation may occur by:

$$CaFe_2O_4 + 1/3CO + 2/3CO_2 \rightarrow 2/3Fe_3O_4 + CaCO_3; \qquad (9)$$

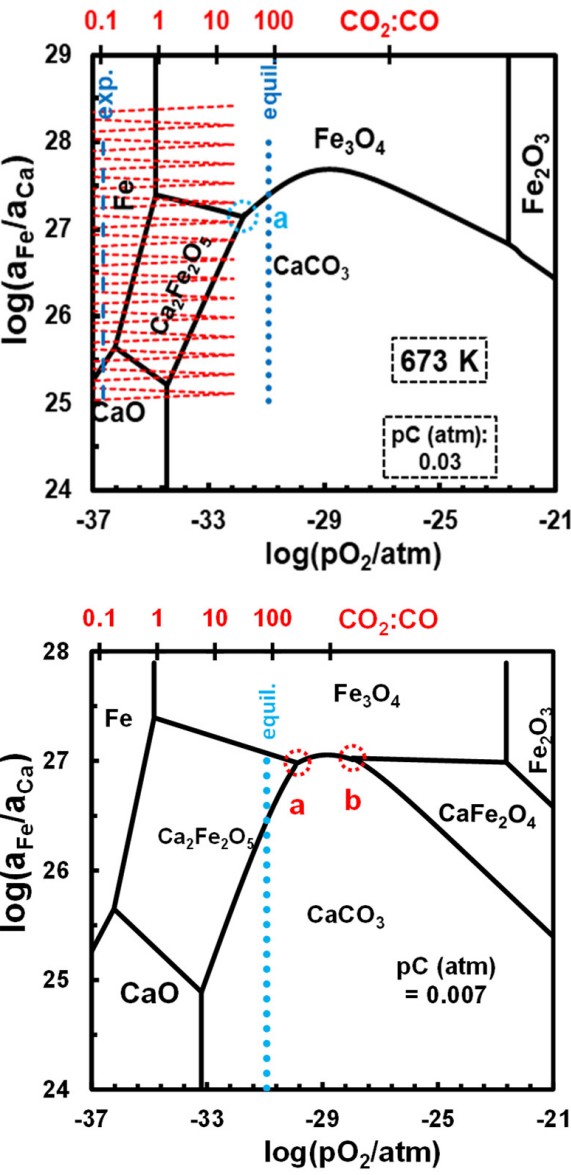

**Figure 8.** Phase stability for the Ca−Fe−O−C diagram vs. oxygen partial pressure, at 400 °C for $p_C = pCO + pCO_2 \approx 0.03$ *atm* (top) and $\approx 0.007$ *atm* (bottom). The secondary horizontal axis shows the corresponding scale for the $CO:CO_2$ ratio. The shaded area shows the redox range that should give rise to deposition of carbon in equilibrium with the gas mixture. The vertical blue lines show the expected redox condition for a feed ratio $H_2O:CO = 2$ in equilibrium (dotted line) or based on the experimental results for the $CO_2:CO$ ratio (dashed line).

Figure 9 shows the equilibrium phase diagram predicted for the Ca−Fe−O−C system at 450 °C, superimposed on the expected redox conditions of the reacting gas mixture with $p_C = pCO + pCO_2 \approx 0.03$ atm and feed ratio $H_2O:CO = 2$. In this case, the phase stability range of the $Ca_2Fe_2O_5$ comprised the effective experimental results for the $CO_2:CO$ ratio (dashed vertical line) and also extended to the predicted equilibrium conditions for the reacting gas mixture (dotted vertical line). For less-reducing conditions, one finds the 3-phase point $Ca_2Fe_2O_5/CaFe_2O_4/Fe_3O_4$ (**a** in Figure 9), and one might assume that

this provides oxygen storage ability to assist conversion of CO, possibly combined with adsorption of $CO_2$ on the catalyst surface, as indicated by FTIR spectra (Figure 5):

$$2CaFe_2O_4 + 1/3CO \rightarrow Ca_2Fe_2O_5 + 2/3Fe_3O_4 + 1/3CO_2(\text{ads}). \tag{10}$$

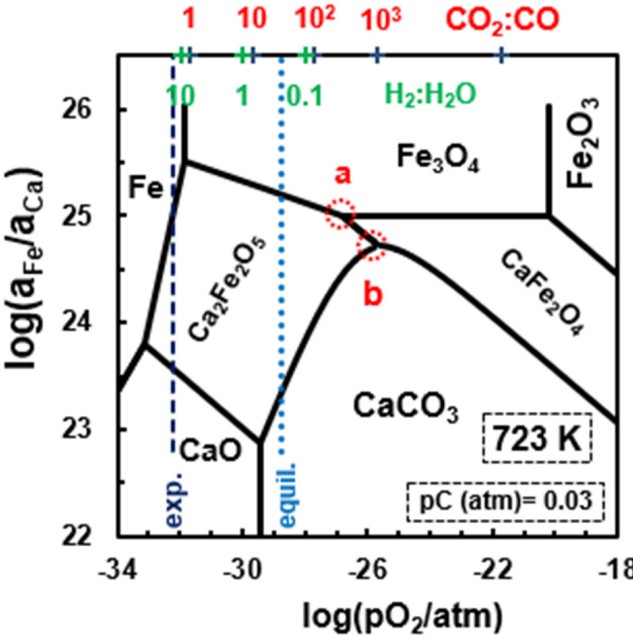

**Figure 9.** Phase stability for the Ca−Fe−O−C diagram vs. oxygen partial pressure, at 450 °C for $p_C = pCO + pCO_2 \approx 0.03$ atm. The vertical blue line shows the expected redox condition for a feed ratio $H_2O$:CO = 2 in equilibrium (dotted line) or based on the experimental results for the $CO_2$:CO ratio (dashed line).

Prospects to promote CO oxidation near the ternary $Ca_2Fe_2O_5$/$CaFe_2O_4$/$Fe_3O_4$ point may also provide clues about the microwave boosting of catalytic performance, based on selective microwave heating of $Fe_3O_4$ [43]; this is consistent with the increasing contents of $Fe_3O_4$ after catalytic testing at temperatures above 400 °C (Figure 4), by reduction of the residual content of $Fe_2O_3$ in the as-prepared catalyst and subsequent segregation of magnetite.

The phase diagram at 450 °C also predicts the 3-phase contact $Ca_2Fe_2O_5$/$CaFe_2O_4$/$CaCO_3$ (**b** in Figure 9), which may also promote subsequent carbonation:

$$Ca_2Fe_2O_5 + CO_2 \rightarrow CaFe_2O_4 + CaCO_3; \tag{11}$$

Figure 10 shows the phase stability of the Ca−Fe−O−C system at 500 °C, superimposed on the relevant gas phase conditions predicted for $pCO + pCO_2 \approx 0.03$ atm and feed ratio $H_2O$:CO = 2. In this case, oxidation of CO to $CO_2$ might also be promoted near the 3-phase contact $Ca_2Fe_2O_5$/$CaFe_2O_4$/$Fe_3O_4$ (**a** in Figure 10), as mentioned for catalytic tests at 450 °C (Equation (10)). In addition, one may assume oxidation of CO to $CO_2$ by slight oxygen deficiency of $Ca_2Fe_2O_5$, combined with significant oxygen permeability [24,44], as follows:

$$Ca_2Fe_2O_5 + \delta CO \rightarrow Ca_2Fe_2O_{5-\delta} + \delta CO_2 \tag{12}$$

$Ca_2Fe_2O_5$ is also known for its sensitivity to CO and $CO_2$ [45], even if the actual mechanisms of CO oxidation might be somewhat complex, as emphasized by DFT simulation, which identified different controlling steps such as oxygen diffusion [46] or formation of $2CO_2$—$Ca_2Fe_2O_{5-\delta}$ complexes [47].

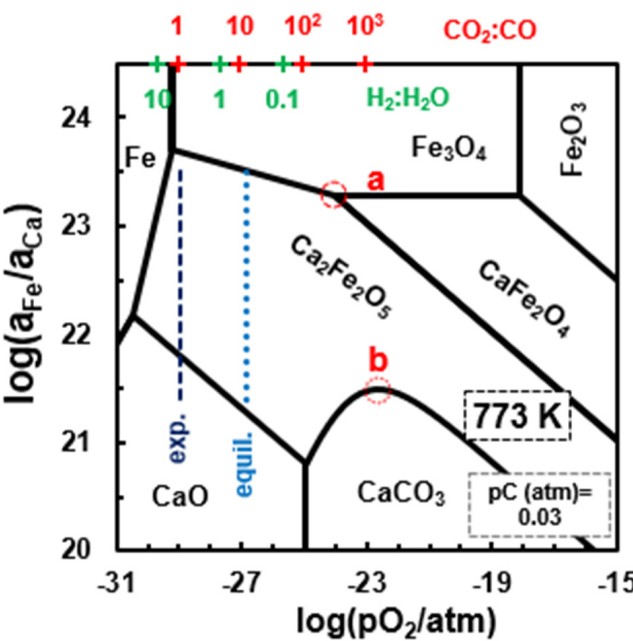

**Figure 10.** Phase stability for the Ca−Fe−O−C diagram vs. oxygen partial pressure, at 500 °C for $p_C = pCO + pCO_2 \approx 0.03\ atm$, and feed ratio $H_2O{:}CO = 2$. The dotted vertical line shows the equilibrium condition for the gas mixtures, and the dashed vertical line the experimental results for the $CO_2{:}CO$ ratio.

Onset of calcium carbonate at 500 °C must also rely on the point defect chemistry of brownmillerites rather than 3-phase contacts. In fact, anti-Frenkel and/or partial Schottky defects might allow significant concentrations of oxygen ion ($V_O^{\bullet\bullet}$) and A-site vacancies ($V_A''$) [48], with corresponding deviations from ideal stoichiometry, $A_{2-x}Fe_2O_{5-\delta}$, where A = Sr, Ca. Thus, one may assume carbonation along the 2-phase boundary $Ca_2Fe_2O_5/CaCO_3$ (**b** in Figure 10), as follows:

$$Ca_2Fe_2O_5 + \delta CO_2 \rightarrow Ca_{2-\delta}Fe_2O_{5-\delta} + \delta CaCO_3. \tag{13}$$

### 2.4. Prospective Applicability to Upgrade Producer Gases

Table 2 summarizes the actual WGS results with and without $Ca_2Fe_2O_5$-based catalyst, with conventional electric heating or under microwave irradiation, and literature data on some of the most relevant WGS catalysts. The present results obtained at 500 °C under microwave irradiation were close to those reported for some established high-temperature (Fe-Cr)-based catalysts [4] but quite lower than those reported for other optimized catalysts, such as Ca-Fe-Ox [49] or Pt-NaA zeolites [50]. Still, improvements are expected by subsequent optimization of the $Ca_2Fe_2O_5$-based catalyst, possibly by slight composition changes, as shown for other Fe-based catalysts [7] or optimized processing. Other (Cu-Zn)-based catalysts perform even better at lower temperatures (≤300 °C) [30]. However, these relatively low temperatures imply greater risks of condensation of tars, which are often found in producer gases. In addition, still-higher temperatures (>500 °C) might allow further increases in CO conversion and yield of $H_2$, as demonstrated with siderite–concrete composite catalysts [51]. Note that the authors also used a representative composition of producer gas to confirm its upgrading with enhanced conversion of CO and increased yield of $H_2$ at up to 700 °C. In this temperature range, one may even seek a contribution of Fe-based catalysts to promote tar conversion, possibly relying on additions of other elements, such as Ni [52]. Sulphur tolerance is also expected based on the demonstrated ability of calcium ferrites to capture $H_2S$ [23], and also taking into account guidelines for other Fe-based catalysts [2,53].

**Table 2.** Comparison of relevant WGS catalytic results with corresponding results from literature sources.

| T (°C) | H$_2$O:CO | GHSV (h$^{-1}$) | Catalyst | Heating | X$_{CO}$ (%) | H$_2$ Yield (%) | Ref. |
|---|---|---|---|---|---|---|---|
| 400 | | | | electric | | ≈49 | |
| 500 | | | | electric | | ≈50 | |
| 400 | 1:1 | | | | | ≈44 | |
| 500 | | ≈28,000 | (Fe-Cr)-based | | | ≈58 | [4] |
| 400 | 2:1 | | | microwave | ≈49 | | |
| 400 | 4:1 | | | | ≈67 | | |
| 500 | 2:1 | | | | ≈71 | | |
| 200 | 2:1 | | | | ≈42 | | |
| 300 | 2:1 | ≈28,000 | (Cu-Zn)-based | microwave | ≈62 | | [30] |
| 300 | 4:1 | | | | ≈93 | | |
| 400 | 1.25:1 | ≈6000 | Ce-Fe-Ox | electric | ≈36–46 | | [49] |
| 500 | | | | | ≈71–84 | | |
| 300 | 2:1 | ≈9000 | Pt-NaA zeolite | electric | ≈15 | | [50] |
| 400 | | | | | ≈68 | | |
| 500 | 2:1 | 14,500–18,250 | 75%sid. + 25%conc. | | 19 | | |
| 600 | 3:1 | 14,500–18,250 | 75%sid. + 25%conc. | electric | 47 | | [51] |
| 500 | 3:1 | 14,500–18,250 | 50%sid. + 50%conc. | | 48 | | |
| 600 | 1:1 | 14,500–18,250 | 50%sid. + 50%conc. | | 40 | | |
| 400 | 2:1 | | | | 10 | 6 | |
| 500 | 2:1 | | | microwave | 52 | 51 | |
| 500 | 3:1 | ≈6000 | Ca$_2$Fe$_2$O$_5$ | | 66 | 58 | present work |
| 500 | 3:1 | | | electric | 24 | 16 | |
| 500 | 3:1 | | blank | microwave | 9 | 0.2 | |

## 3. Materials and Methods

### 3.1. Catalyst Preparation

Ca$_2$Fe$_2$O$_5$-based pellets were prepared by reactive firing of low-cost natural siderite (SIDCO Minerals Inc., Linden, TX, USA) and calcite (CaCO$_3$, LabChem, Tokyo, Japan), based on a method reported earlier [20]. The X-ray diffraction of natural siderite showed FeCO$_3$ as the main crystalline phase, combined with quartz (SiO$_2$). However, further chemical analysis by XRF spectrometry (Philips PW 1400/00, Philips, Amsterdam, The Netherlands) also revealed the presence of significant fractions of aluminum and alkaline earth elements (Ca and Mg), and traces of Mn and Ti. Assuming that Fe, Ca and Mg are present as carbonates, and that the remaining elements are in oxide form, the estimated elemental composition of the siderite is 80.8% Fe, 10.8% Si, 6.2% Al, 0.9% Mg, 0.9% Ca, 0.2% Mn and 0.2% Ti.

Stoichiometric mixtures of CaCO$_3$ and siderite (based on the elemental fraction of Fe) were ball milled (Retsch PM 100, Retsch, Berlin, Germany) at 400 rpm for 3 h, using a Teflon vial (~375 cm$^3$) and zirconia balls (TOSOH Co.) with diameters of 1.5 cm and 1 cm, in the proportion of 10 and 40, respectively. The ball to powder weight ratio was ~2.5:1, and undue heating was avoided by milling for periods of 5 min with a subsequent pause of 2 min. The quantity of activated powder prepared in each milling experiment was ~60 g. The mechanically activated precursor mixture was then used to process bar-shaped catalyst

samples (~0.7 × 0.3 × 0.3 cm) by cold isostatic pressure at 200 MPa, with subsequent thermochemical treatment in air at 800 °C for 4 h, with a heating rate of 2 °C/min.

### 3.2. Water Gas Shift (WGS) Tests

Testing of the WGS reaction was performed with and without catalyst (blank) using a 1.8 kW microwave system (PYRO Microwave Furnace, Milestone Srl., Milan, Italy) that contains a fixed-bed reactor, as shown in Figure 11. The microwave irradiation was performed through an industrial magnetron, connected to a power supply with 4 kV high tension and filament current. The system also incorporated an evaporator chamber, externally heated by a heater tape. Each experiment was performed following the same protocol: 15 g of catalyst supported on two layers of ceramic wool was loaded into the fixed-bed reactor and heated at 5 °C·min$^{-1}$ in N$_2$ atmosphere to the reaction temperature. Afterwards, H$_2$O was injected into the evaporator by an HPLC pump (Jasco, PV-980 model, Tokyo, Japan) and carried by a gas mixture (0.5 L$_{STP}$·min$^{-1}$) containing 10 vol.% CO and N$_2$ (balance) into the reactor. The gas mixture was fed by a mass flow controller (MFC, Alicat, MCS Series, Alicat Scientific, Thane, India), whereas the temperature of the catalyst bed was controlled by an infrared sensor located on the right side of the unit. To prevent condensation, all lines were maintained at 250 °C through the use of heating tapes. At the reactor outlet, the gas product passed through a set of traps, which removed the unconverted water by condensation, before being collected in sample bags for off-line analysis by gas chromatography (Micro-GC Fusion, INFICON, Bad Ragaz, Switzerland, equipped with wall-coated open tubular (WCOT) and porous layer open tubular (PLOT) GC columns). The tests were performed at temperatures in the range 400–500 °C (microwave power from 410 to 520 W), with steam to carbon monoxide feed ratio (H$_2$O:CO) in the range 2:1 to 4:1. The gas hourly space velocity (GHSV) changed from 5900–6750 h$^{-1}$ due to steam addition. Some tests were also performed with a conventional electric furnace for comparison.

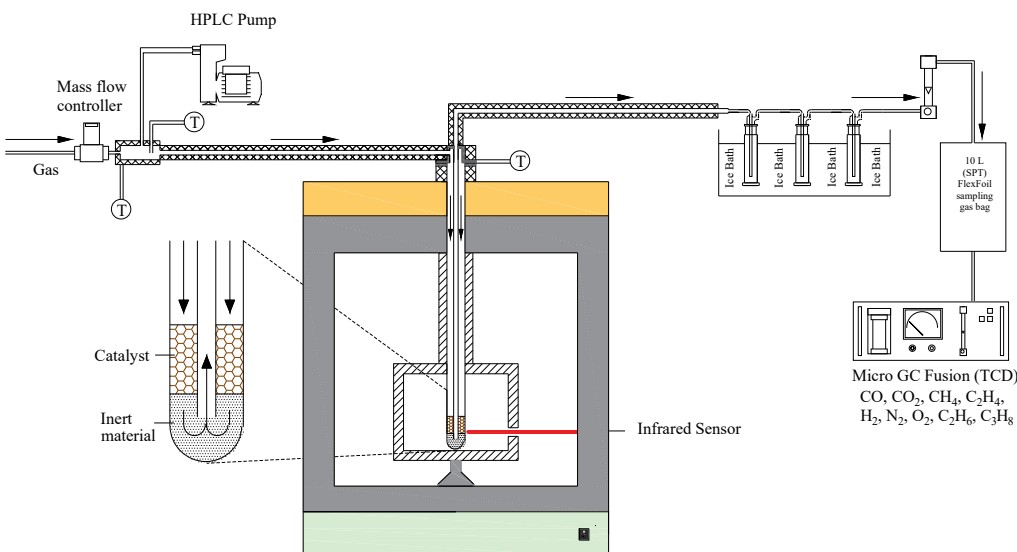

**Figure 11.** Schematic representation of the experimental apparatus used for water gas shift reaction tests in a microwave chamber.

The performance of the WGS reaction was evaluated, by the conversion of CO and through H$_2$ and CO$_2$ yields, relative to the total contents of C-containing species in the gas mixture, at the reactor outlet, i.e.,

$$Y_{H_2} = \frac{\dot{n}H_2}{\dot{n}CO + \dot{n}CO_2 + \dot{n}CH_4 + \cdots} \tag{14}$$

$$Y_{CO_2} = \frac{\dot{n}CO_2}{\dot{n}CO + \dot{n}CO_2 + \dot{n}CH_4 + \cdots} \tag{15}$$

where $\dot{n}H_2$, $\dot{n}CO_2$, $\dot{n}CO$, $\dot{n}CH_4+$ are the molar flow rates of $H_2$, $CO_2$, $CO$, $CH_4 \ldots$ (mol·min$^{-1}$) at the reactor outlet, respectively; this was best-suited to minimize the impact of experimental errors in critical steps such as injection of $H_2O$ in the evaporator or collection of samples for gas chromatography.

### 3.3. Catalyst Characterization

X-ray diffractograms of the as-prepared catalyst and spent catalyst were carried out using a PANalytical X'Pert PRO[3] diffractometer equipped with graphite monochromator (CuK$_\alpha$ radiation, scan step size = 0.02°) to monitor phase changes. Additionally, postmortem analysis of the catalyst samples after WGS experiments was performed by Fourier-transform infrared spectroscopy (FTIR, Thermo Fisher Scientific, Waltham, MA, USA). The spectra of the samples were recorded by accumulating 64 scans at 4 cm$^{-1}$ resolution in the spectral range of 400–4000 cm$^{-1}$ using a Galaxy Series FT-IR 700 spectrometer equipped with a DTGS CsI detector. A scanning electron microscope (SEM, Hitachi, TM4000 Plus, Tokyo, Japan) equipped with energy dispersive X-ray spectroscopy (EDS Oxford Inca TEM250, Oxford Instruments, Abingdon, UK) was used for microstructural characterization and to assess eventual changes under the conditions of catalytic testing.

Quasi planar phase stability diagrams were derived for the Ca-Fe-O and Ca-Fe-O-C systems in representative redox conditions and represented as a function of the activity ratio $a_{Fe} : a_{Ca}$ in the condensed phases, and oxygen partial pressure in the gas phase ($pO_2$). Diagrams for the Ca-Fe-O system were derived by a method detailed elsewhere [54,55], and the extension for the Ca-Fe-O-C system is described in Appendix A.

## 4. Conclusions

Low-cost $Ca_2Fe_2O_5$-based catalysts prepared from natural mineral precursors catalyzed the WGS reaction while hindering the side reaction of methanation and preventing carbon deposition; this confirmed prospects to enhance the $H_2$ yield of producer gas. The catalytic performance achieved under microwave heating was also much better than by conventional electric heating. Post-mortem analyses of spent catalyst samples by XRD and FTIR provided evidence to interpret the catalytic activity of the actual $Ca_2Fe_2O_5$-based catalysts, taking into account the onset of $Fe_3O_4$ and calcium carbonate polymorphs, combined with adsorbed $CO_2$. Phase diagrams predicted for the Ca-Fe-O-C system also provided relevant guidelines for the mechanisms of CO oxidation, carbonation and onset of $Fe_3O_4$, in close relation with 3-phase contacts, or relying on the point defect chemistry of the brownmillerite phase; this allows variable oxygen stoichiometry by redox cycles, probably combined with deviations from ideal stoichiometry of the brownmillerite phase (Ca:Fe < 1). Ready onset of $Fe_3O_4$ under the redox conditions of WGS may also explain high prospects for microwave boosting, based on the superior magnetic properties of $Fe_3O_4$. Still, further work is required to reach a comprehensive understanding of the role of microwaves in catalytic performance.

**Author Contributions:** Conceptualization, I.A., L.C.M.R. and J.R.F.; Methodology, I.A., L.C.M.R. and J.R.F.; Validation, J.R.F.; Formal analysis, I.A. and J.R.F.; Investigation, I.A. and L.C.M.R.; Resources, L.A.C.T. and J.R.F.; Writing—original draft preparation, J.R.F.; Writing—review & editing, I.A., L.C.M.R., L.A.C.T. and J.R.F.; Supervision, J.R.F.; Project administration, L.A.C.T.; Funding acquisition, L.A.C.T. and J.R.F. All authors have read and agreed to the published version of the manuscript.

**Funding:** This research was funded through project NOTARGAS (ref. POCI-01-0145-FEDER-030661), CESAM–Centre for Environmental and Marine Studies (UIDB/50017/2020, UIDP/50017/2020, & LA/P/0094/2020), and CICECO–Aveiro Institute of Materials (UIDB/50011/2020, UIDP/50011/2020 & LA/P/0006/2020), financed by national funds through Portuguese Foundation for Science and Technology (FCT)/Ministry of Science, Technology and Higher Education (MCTES).

**Data Availability Statement:** Data are contained within the article.

**Conflicts of Interest:** The authors declare no conflict of interest.

**Appendix A**

Diagrams for the Ca-Fe-O system were computed from free-energy calculations of 2-solid phase equilibria, at constant temperature. Representative cases of 2-solid phase reactions may depend on $pO_2$ only (Equation (A1)), on the activity ratio $a_{Fe} : a_{Ca}$ (Equation (A2)), or on both (Equation (A3)):

$$6FeO + O_2 \overset{k_1}{\Leftrightarrow} 2Fe_3O_4;$$

$$log(pO_2) = \frac{\Delta G_1}{2.30RT} \tag{A1}$$

$$Ca + Fe_3O_4 \overset{k_2}{\Leftrightarrow} Fe + CaFe_2O_4;$$

$$log\left(\frac{a_{Fe}}{a_{Ca}}\right) = -\frac{\Delta G_2}{2.30RT} \tag{A2}$$

$$Ca + \frac{2}{3}Fe_3O_4 \overset{k_3}{\Leftrightarrow} Fe + 0.5Ca_2Fe_2O_5 + \frac{1}{12}O_2;$$

$$log\left(\frac{a_{Fe}}{a_{Ca}}\right) = -\frac{\Delta G_3}{2.30RT} - \frac{1}{12}log(pO_2) \tag{A3}$$

Onset of $CaCO_3$ by reaction of CaO with atmospheric $CO_2$ (Equation (A4)) in the redox conditions of gasification can be coupled with $CO/CO_2$ equilibrium (Equation (A5)), to establish the redox dependence:

$$CaO + CO_2 \overset{k_4}{\Leftrightarrow} CaCO_3$$

$$pCO_2 = 1/k_4 \tag{A4}$$

$$CO + 0.5O_2 \overset{k_5}{\Leftrightarrow} CO_2 \tag{A5}$$

$$\left(\frac{pCO_2}{pCO}\right) = k_5 pO_2^{1/2}$$

In addition, one must consider another specific relation between these gases, such as the overall content $pCO + pCO_2 \approx p_C$ determined by the chemical composition of the biomass and the relation between the gasification agent and biomass; this yields:

$$pCO_2 = \frac{p_C}{1 + \left(k_5 pO_2^{1/2}\right)^{-1}} \tag{A6}$$

$$pCO = \frac{p_C}{1 + k_5 pO_2^{1/2}} \tag{A7}$$

Thus, on combining with Equation (A4), one obtains:

$$log(pO_2) = -2log\{k_5(k_4 p_C - 1)\} \tag{A8}$$

Redox conditions for onset of carbon can be described by disproportionation of carbon monoxide, combined with Equations (A6) and (A7):

$$2CO \overset{k_9}{\Leftrightarrow} C + CO_2$$

$$k_9 = \frac{pCO_2}{(pCO)^2} = \frac{\left(1 + k_5 pO_2^{1/2}\right)k_5 pO_2^{1/2}}{p_C} \tag{A9}$$

Redox conditions for the onset of $Fe_3C$ may be predicted for reactions of CO with metallic Fe or with iron oxides (FeO or $Fe_3O_4$), as shown in Table A1. In addition, Table A1 shows the 2-phase equilibria derived for $CaCO_3$ and mixed oxide phases ($Ca_2Fe_2O_5$ or $CaFe_2O_4$) or between $CaCO_3$ and different iron oxides (FeO, $Fe_3O_4$ or $Fe_2O_3$).

**Table A1.** Relevant equilibria of C-based species in the Ca-Fe-O-C system. The corresponding solid phases are shown in **bold**.

| Reaction | Redox Conditions |
|:---:|:---:|
| $\boldsymbol{CaO} + CO_2 \overset{k_4}{\Leftrightarrow} \boldsymbol{CaCO_3}$ | $log(pO_2) = -2log\{k_5(k_4p_C - 1)\}$ |
| $2CO \overset{k_9}{\Leftrightarrow} \boldsymbol{C} + CO_2$ | $k_9 = \dfrac{(1+k_5pO_2^{1/2})k_5pO_2^{1/2}}{p_C}$ |
| $3\boldsymbol{Fe} + CO \overset{k_{10}}{\Leftrightarrow} \boldsymbol{Fe_3C} + 0.5O_2$ | $k_{10} = \dfrac{(1+k_5pO_2^{1/2})pO_2^{1/2}}{p_C}$ |
| $3\boldsymbol{FeO} + CO \overset{k_{11}}{\Leftrightarrow} \boldsymbol{Fe_3C} + 2O_2$ | $k_{11} = \dfrac{(1+k_5pO_2^{1/2})pO_2^2}{p_C}$ |
| $\boldsymbol{Fe_3O_4} + CO \overset{k_{12}}{\Leftrightarrow} \boldsymbol{Fe_3C} + 2.5O_2$ | $k_{12} = \dfrac{(1+k_5pO_2^{1/2})pO_2^{2.5}}{p_C}$ |
| $Ca + 2/3\boldsymbol{Fe_3C} + 19/12O_2 \overset{k_{13}}{\Leftrightarrow}$ $Fe + 0.5\boldsymbol{Ca_2Fe_2O_5} + 2/3CO$ | $k_{13} = \left(\dfrac{a_{Fe}}{a_{Ca}}\right)\dfrac{(p_C)^{2/3}}{pO_2^{19/12}(1+k_5pO_2^{1/2})^{2/3}}$ |
| $Ca + 1/3\boldsymbol{Fe_3C} + 2/3O_2 \overset{k_{14}}{\Leftrightarrow} Fe + \boldsymbol{CaO} + 1/3CO$ | $k_{14} = \left(\dfrac{a_{Fe}}{a_{Ca}}\right)\dfrac{(p_C)^{1/3}}{pO_2^{2/3}(1+k_5pO_2^{1/2})^{1/3}}$ |
| $Ca + 0.5\boldsymbol{Ca_2Fe_2O_5} + 2CO + 0.75O_2 \overset{k_{15}}{\Leftrightarrow}$ $Fe + 2\boldsymbol{CaCO_3}$ | $k_{15} = \left(\dfrac{a_{Fe}}{a_{Ca}}\right)\dfrac{(1+k_5pO_2^{1/2})^2}{pO_2^{0.75}(p_C)^2}$ |
| $Ca + 0.5\boldsymbol{CaFe_2O_4} + 1.5CO_2 \overset{k_{16}}{\Leftrightarrow}$ $Fe + 1.5\boldsymbol{CaCO_3} + 0.25O_2$ | $k_{16} = \left(\dfrac{a_{Fe}}{a_{Ca}}\right)\dfrac{\{1+(k_5pO_2^{1/2})^{-1}\}^{1.5}pO_2^{0.25}}{(p_C)^{1.5}}$ |
| $Ca + FeO + CO + 1/2O_2 \overset{k_{17}}{\Leftrightarrow} Fe + \boldsymbol{CaCO_3}$ | $k_{17} = \left(\dfrac{a_{Fe}}{a_{Ca}}\right)\dfrac{1+k_5pO_2^{1/2}}{pO_2^{1/2}p_C}$ |
| $Ca + 1/3\boldsymbol{Fe_3O_4} + CO + 1/3O_2 \overset{k_{18}}{\Leftrightarrow} Fe + \boldsymbol{CaCO_3}$ | $k_{18} = \left(\dfrac{a_{Fe}}{a_{Ca}}\right)\dfrac{1+k_5pO_2^{1/2}}{pO_2^{1/3}p_C}$ |
| $Ca + 0.5\boldsymbol{Fe_2O_3} + CO_2$ $\overset{k_{19}}{\Leftrightarrow} Fe + \boldsymbol{CaCO_3} + 0.25O_2$ | $k_{19} = \left(\dfrac{a_{Fe}}{a_{Ca}}\right)\dfrac{\{1+(k_5pO_2^{1/2})^{-1}\}pO_2^{0.25}}{p_C}$ |

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
