# Peer review of "Ca2Fe2O5-Based WGS Catalysts to Enhance the H2 Yield of Producer Gases"

_catalysts, doi:10.3390/catal14010012_

Round 1

Reviewer 1 Report

Comments and Suggestions for Authors

This paper by Antunes et al. studies the enhancement of H2 yield in WGS reaction over Ca2Fe2O5-based catalysts. Although the results obtained in this study are interesting and valuable for the researchers who devoted in same research field, the points listed below are raised from the reviewer as queries and concerns. So, the reviewer would like to make "major revision" decision for this paper.

Major comments

1.       It is not clear what kind of processes Ca2Fe2O5-based catalysts can be applied to. Producer gas from biomass gasification often has an H2/CO ratio lower than 1, and can be supplied to the Fischer-Tropsch reaction by promoting the WGS reaction. Producer gas already contains CO2 and H2. It is necessary to show that it is effective in increasing H2 using such raw feed gas.

2.       Producer gas from biomass gasification usually contains a small amount of H2S and COS. Although this is a poisonous substance for Ca2Fe2O5-based catalysts, comments should be made regarding its life expectancy and tolerance.

3.       Cu/Zn is a common catalyst for WGS reactions at a lower temperature (230-300℃). Ca2Fe2O5-based catalysts should be compared with Cu/Zn catalysts, and advantages and disadvantages should be added, because producing high-temperature steam (400-500℃) requires a relatively large amount of input energy.

Minor comments

1.       Keywords are missing.

2.       2. Materials and Methods, Definitions of CO2 and H2 yields should be added.

Comments on the Quality of English Language

This paper by Antunes et al. studies the enhancement of H2 yield in WGS reaction over Ca2Fe2O5-based catalysts. Although the results obtained in this study are interesting and valuable for the researchers who devoted in same research field, the points listed below are raised from the reviewer as queries and concerns. So, the reviewer would like to make "major revision" decision for this paper.

Major comments

1.       It is not clear what kind of processes Ca2Fe2O5-based catalysts can be applied to. Producer gas from biomass gasification often has an H2/CO ratio lower than 1, and can be supplied to the Fischer-Tropsch reaction by promoting the WGS reaction. Producer gas already contains CO2 and H2. It is necessary to show that it is effective in increasing H2 using such raw feed gas.

2.       Producer gas from biomass gasification usually contains a small amount of H2S and COS. Although this is a poisonous substance for Ca2Fe2O5-based catalysts, comments should be made regarding its life expectancy and tolerance.

3.       Cu/Zn is a common catalyst for WGS reactions at a lower temperature (230-300℃). Ca2Fe2O5-based catalysts should be compared with Cu/Zn catalysts, and advantages and disadvantages should be added, because producing high-temperature steam (400-500℃) requires a relatively large amount of input energy.

Minor comments

1.       Keywords are missing.

2.       2. Materials and Methods, Definitions of CO2 and H2 yields should be added.

Reviewer 2 Report

Comments and Suggestions for Authors

The manuscript is well organized and discussed with enough experimental data and relevant analysis, which can be accepted for publication.

Reviewer 3 Report

Comments and Suggestions for Authors

General Comment: The authors developed Ca2Fe2O5-based catalysts to evaluate its catalytic performance towards water gas shift reactions. The authors claim and prove that they were able to suppress the side reaction (methanation) and there was no catalyst coking observed, thereby enhancing H2 yield under microwave irradiation. Further analysis of the reused catalyst showed that the catalyst was cable of adsorbing CO2 to form CaCO3. In addition, in-depth study using phase diagrams provide insights on the mechanisms of CO oxidation, carbonation and onset of Fe3O4. Therefore, I suggest acceptance of this manuscript after major revision as follows:

1)     It is very important that the authors describe their previous work in the introduction and compare the results from their previous studies in the results and discussion sections.

2)     The introduction part is suggested to be well-organized to show the novelty of this work.

3)     The authors should particularly represent the amount (both mmol and ml/mg) of all the catalysts that were loaded into the reactor? In addition, were the catalysts diluted with quartz before loading in the reactor?

4)     The authors must provide the make and specifications of the microwave chamber. How much heating (in Watts) was used to perform the microwave assisted reactions?

5)     The authors are requested to mention the details of the steam generator? How exactly was water injected into the reactor? In traditional cases, water is usually pumped through a syringe pump ensuring that the heating lines are kept in the range of 120-150 °C. Please clarify.

6)     At what partial pressures was the CO and H2O reactants varied? Were these reactions performed under differential conditions?

7)     The authors should specify the details of the GC and the GC column configurations.

8)     Hoe did the authors confirm the Ca2Fe2O5 shows a bar-shaped morphology. The authors must provide FE-Sem or HR-TEM images to support their claim.

9)     Page 3, Line 96: Do the authors mean to write “blank”? If so, look for the spelling mistake.

10)  In Figure 2, the authors must mention the JCPDS/ PDF no. corresponding to all the materials. In addition, it will be beneficial to the readers if the crystallite size is calculated using Debye-Scherrer equation.

11)  Page 4, Line 160: How were the intensity ratios calculated?

12)  What is the surface area and pore diameter of the catalyst prepared in this work. This is necessary because it is essential to know the exposure of catalytically active sites based on their surface area.

13)  In addition, I would also recommend the authors to perform XPS analysis of the catalyst to get the elemental composition and the oxidation state of the catalyst.

14)  I strongly recommend to add a comparison table with other literature reports for different catalysts used for WGS catalysts under microwave conditions.

15)  The authors are requested to check the manuscript thoroughly for grammatical and typographical errors. Please polish the whole manuscript carefully. Some small mistakes such as units should be checked in all the manuscript.

Comments on the Quality of English Language

 Minor editing of English language required

Round 2

Reviewer 1 Report

Comments and Suggestions for Authors

The authors addressed to my review comments, and the revised manuscript is suitable for publication.

Reviewer 3 Report

Comments and Suggestions for Authors

I am satisfied with the responses to my comments. The authors have addressed all the comments and enhanced the quality of the manuscript for better readership. The manuscript can be accepted in its current form.

Comments on the Quality of English Language

Minor editing of English language required